# Synthetic 3D Recording of a Shipwreck Embedded in Seafloor Sediments: Distinguishing Internal Details

**Lars O. Boldreel** [1,*], **Ole Grøn** [1] **and Deborah Cvikel** [2]

[1] Department of Geosciences and Natural Resource Management, University of Copenhagen, Geology Section, Østervoldgade 10, DK 1250 Copenhagen K, Denmark; og@ign.ku.dk

[2] Leon Recanati Institute for Maritime Studies and Department of Maritime Civilizations, University of Haifa, Haifa 3498838, Israel; dcvikel@research.haifa.ac.il

\* Correspondence: lob@ign.ku.dk

**Abstract:** 3D recording of shipwrecks completely buried in seafloor sediments has great potential as an important aspect of maritime archaeological surveys and management. Buried shipwrecks have been recorded directly with seismic 3D Chirp sub-bottom profilers on an experimental basis. This method is, however, expensive, time-consuming and complicated. This article outlines the application of a faster, cheaper, and less complicated method of synthetic 3D recording, which is also less sensitive to weather conditions. It involves the acquisition of a larger number of seismic 2D high-resolution sub-bottom profiles in a dense grid that does not need to be regular. The method is based on the results of survey work conducted in the Akko Harbour area, on the Carmel coast of Israel, which shows that the shape of the hull of a shipwreck can be precisely determined, and that the sedimentary units bounding it can be outlined and interpreted. Based on an interpretation of the shape of the hull, the depth of the structure was measured, and a 3D image of the shipwreck was subsequently generated. Samples of the sub-seafloor were obtained across the area, and the sample located within the area of the mapped shipwreck was found to contain wood fragments and a piece of rope. This article demonstrates that 2D surveying is a viable and cost-effective alternative to 3D surveying that is able to produce good results.

**Keywords:** Akko 4; Chirp; detection; mapping; maritime archaeology; shipwreck; sub-bottom seismics; wave compensation

## 1. Introduction

Acoustic systems, such as side-scan and multibeam sonars and sub-bottom profilers are non-destructive seismic techniques well suited to maritime archaeological surveys, which have developed a tradition for their application. These high-frequency systems can detect archaeological objects that are visible above the seafloor surface, but they are unable to detect completely buried objects due to their very limited ability to penetrate seafloor sediments e.g., [1]. This is unfortunate, as a large proportion, possibly even most, of the submerged cultural heritage, for example shipwrecks and Stone Age sites, lies completely buried within these sediments [2–4]. Seismic 3D sub-bottom systems that can penetrate seafloor sediments have been developed, but these are rarely used, as they are relatively expensive, complicated to operate, and the resulting data are difficult to interpret [5–8]. A cheaper and less complicated method of precise detection and imaging of completely buried archaeological objects is to develop synthetic 3D images based on high-resolution 2D seismic sub-bottom profiles e.g., [5,8–15]. Furthermore, in cases where archaeological objects are partially exposed above the seafloor surface, synthetic 3D images based on high-resolution 2D sub-bottom data can provide valuable detailed information on their buried parts.

In relation to shipwrecks, the use of modern Chirp sub-bottom profilers has improved the quality of the 2D sub-bottom data to such a degree that in some cases it is possible to

distinguish their internal structural details, as well as to obtain a detailed understanding of the sedimentation inside and around the wrecks. There is, however, a problem in that the data produced by the better Chirp systems contain so much high-resolution detail that they are considerably more difficult to interpret than the rather simple data from multibeam and side-scan sonar. The maritime archaeological use of Chirp data consequently requires comprehensive training in the interpretation of both small-scale archaeological features and small-scale geological/geophysical phenomena, i.e., additional qualifications to those required for the generally larger-scale interpretation of sub-bottom profiles for purely geological purposes. A positive aspect is that these geo-archaeological details are in most cases extremely useful for an understanding of the relevant archaeological features, and facilitate the development of cost-effective investigation or management and/or conservation strategies. An important element of such activities in the future will be the specific training of interpreters for this type of data.

Exact positioning of the recording lines is an imperative requirement for precise synthetic 3D reconstruction of objects buried in the seafloor based on 2D profiles. Our field experience shows that a stated precision of, for example ±1 m (i.e., 'sub-metre'), can actually be considerably greater in practice [16,17]. A deviation from precise values of this magnitude can lead to significant distortion of the shape of a shipwreck. To avoid such problems, a positioning precision of no greater than ±10 cm is necessary. In our case, precision of this order was obtained by using a standard sub-metre precision Differential Global Positioning System (DGPS) with C-Nav calibration applied. This allows reasonably precise recording of the details of shipwrecks, for example.

The present paper focuses on the 3D reconstruction of the now buried Akko 4 shipwreck in the Akko Harbour area, Israel, from 2014 to 2017. The intense survey activity during this period resulted in the recording of a multitude of profiles covering the central parts of the historic and prehistoric harbour area and, consequently, also of the Akko 4 wreck. The latter therefore became a logical subject for further analysis and experimentation with respect to the level of detail with which it is possible to generate 3D imaging of buried shipwrecks based on 2D data recorded with off-the-shelf seismic equipment.

## 2. The Akko 4 Shipwreck and Its Context

The walled port and city of Akko (Acre, St. Jean d'Acre, Akka) located at the northern extremity of Haifa Bay, northern Israel (Figure 1), has a continuous settlement history with important naval and trade functions from the Early Bronze Age to the Modern Era [18]; [19,20]. It was conquered by the Ottomans in 1516 [20] and, being considered a strategic key location in relation to the Holy Land and Syria, several naval campaigns involving both local and European armies and navies took place in the waters off Akko in the 18th and 19th centuries [21]. Parallel to this, Akko Harbour was used for commercial purposes, as well as scientific expeditions to the Holy Land. Ships of various types and from various fleets—European, eastern Mediterranean, and even American, made use of Akko Harbour, and the Akko 4 wreck is the remains of one of those ships.

An underwater survey conducted by the Israel Antiquities Authority (IAA) in 1990, prior to the construction of a marina in the Akko area, led to the discovery of the Akko 4 shipwreck [22,23]. A remote sensing survey undertaken by the Israel Oceanographic and Limnological Research Institute (IOLR) combined magnetic and acoustic remote sensing by employing a Geometric 880 proton magnetometer and an ORE (Ocean Research Equipment) 3.5-kHz sub-bottom profiler. In 1991, the area was covered by East–West profiles recorded 10 m apart. Positioning during the survey was undertaken with a Motorola mini-ranger installed on the survey boat, utilizing two land stations. The results from the proton magnetometer were poor due to the presence of large amounts of scrap iron in the area. Consequently, most (20) of the potential archaeological anomalies were detected with the sub-bottom profiler (Figure 2), and subsequently checked by diver archaeologists. Of the 20 anomalies recorded, eight proved to be shipwrecks partly embedded in the seafloor sediments and partly exposed and visible [22].

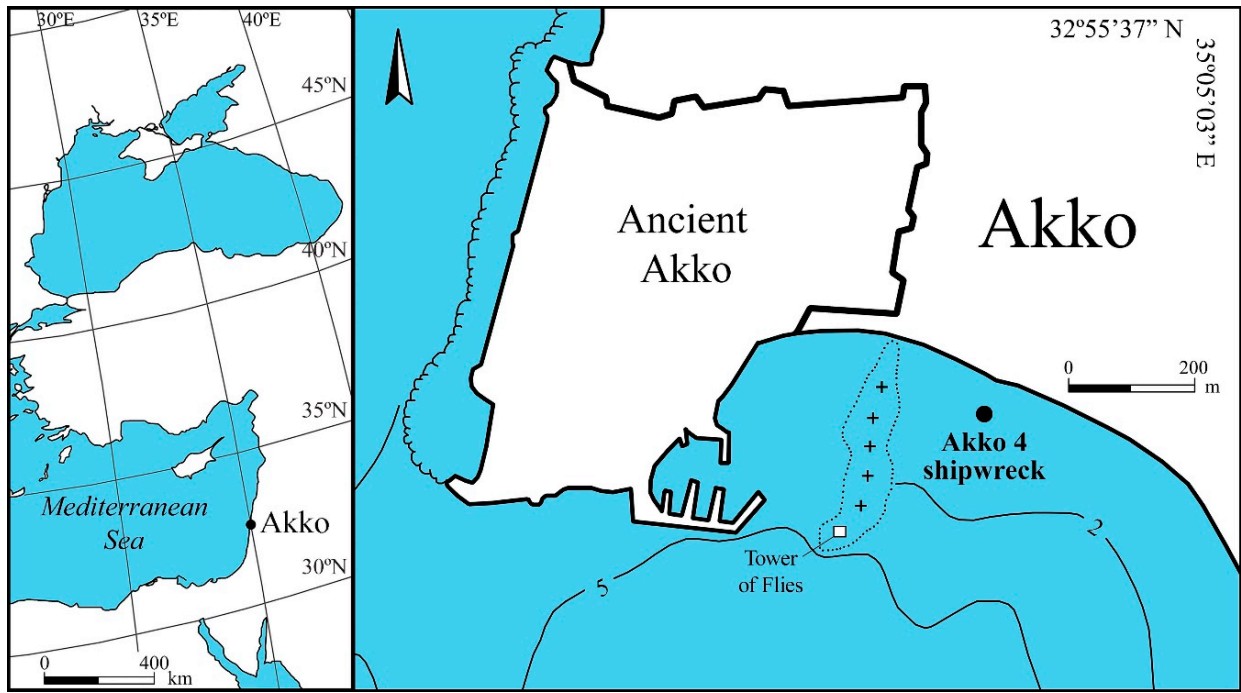

**Figure 1.** The location of Akko/Acre, Israel. Graphics: S. Haad.

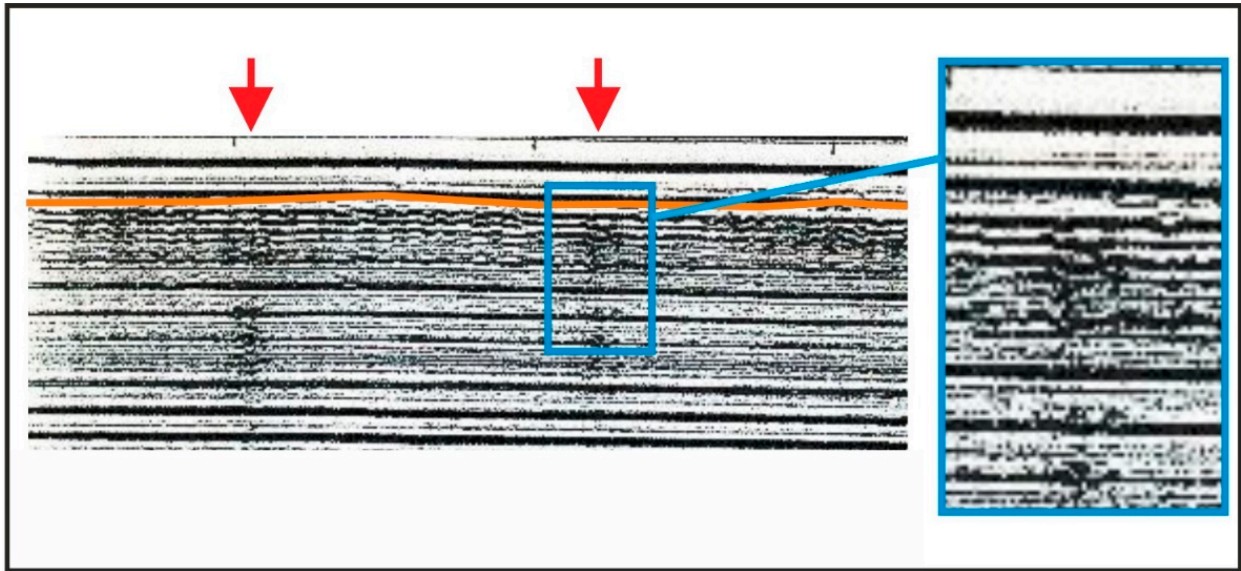

**Figure 2.** Recording from 1990 of one of the shipwrecks at Akko detected below the seafloor with a 3.5-kHz sub-bottom profiler. The red arrows mark two passes above the wreck. The seafloor is shown as an orange line. One of the recordings of the wreck has been enlarged. It was not noted which of the sediment-embedded wrecks this recording shows. Graphics modified by Grøn after [24], courtesy of the Israel Oceanographic and Limnological Research Institute (IOLR) and the Israel Antiquities Authority (IAA).

The Akko 4 shipwreck was discovered about 100 m from the shore in Akko Harbour. The water was about 2-m deep and the wreck lay embedded about 1.5 m into the sand of the seafloor. It was recorded as being 23-m long and 7-m wide. A small trial excavation employing hand-fanning exposed hull planks, framing timbers, and ceiling planks among the hull remains (Figure 3). The ship's fastenings were apparently made of bronze, and the hull had metal sheathing. Two decorated clay tobacco pipes were dated to the 15th–18th

centuries. The shipwreck also contained potsherds, animal bones—probably sheep/goat—and a considerable quantity of nut shells and pulses. The ship was dated to the Late Ottoman period, 18th–19th centuries, based on a single radiocarbon date (RT 1427) [24].

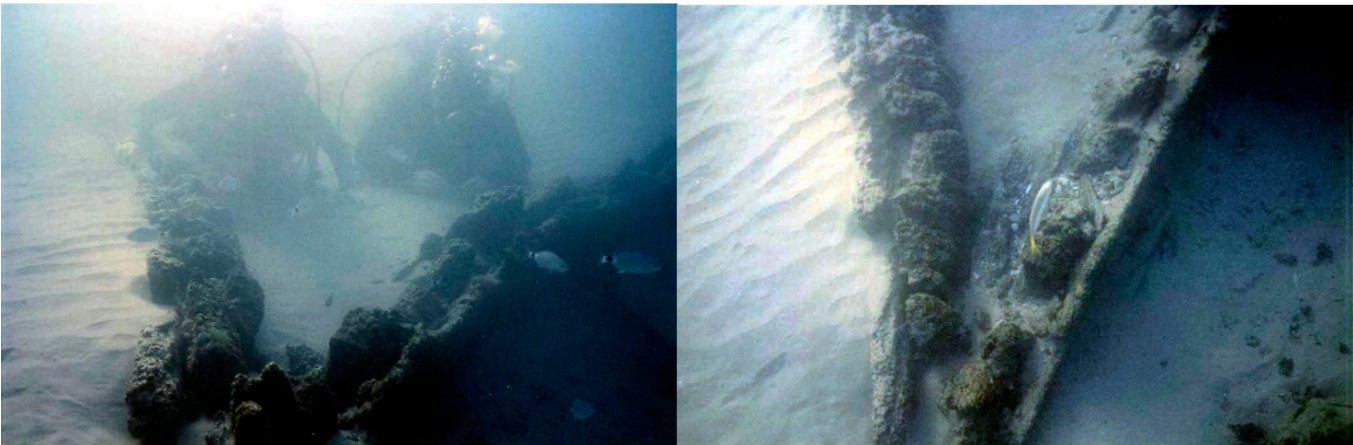

**Figure 3.** The upper part of the Akko 4 shipwreck as seen during the 1991 trial excavation. Photos courtesy of E. Galili, Israel Antiquities Authority.

Since the 1990 survey, the shipwreck has become completely covered by about 40 cm of sand, and therefore a limited underwater survey was conducted in March 2013 to relocate it (IAA survey permit S-399/2013). Water-jetting exposed three timbers of different species (*Populus alba/Populus nigra* (white/black poplar), *Pinus nigra* (black pine), and *Quercus petraea* (sessile oak)), and an 8.5-cm long piece of rope. The three wood samples were AMS (Accelerator mass spectrometry) dated to the calibrated time interval AD 1660–1890 [25], thereby confirming the dating of the ship to the Ottoman period. The wreck is probably the remains of an Ottoman merchantman, but it is impossible to determine the vessel type before a thorough excavation has been carried out.

No substantial excavation was undertaken below the seafloor that could have disturbed the seismic recording. After the 2013 investigation, the shipwreck became completely embedded in naturally deposited sediments, and therefore provided an ideal test case, i.e., a well-documented shipwreck totally embedded in seafloor sediments.

The superficial seafloor sediments around Akko 4 consist of fine quartz sand deposited in water, one to several layers of crushed and intact mollusc shells, and apparently sandy sediments again below these. According to the results of the seismic survey from the harbour area, there are several so far unverified seismic anomalies, which may represent old harbour structures, and which could explain the wooden fragments observed embedded in the seafloor sediments in other parts of the area.

### 3. The 3D Reconstruction of the Akko 4 Shipwreck. An Inner Deck and Ballast Stones?

Detailed information about the hull of the Akko 4 shipwreck and some of its internal features, as well as the process whereby it became infilled with sediment, was elucidated by way of several recorded profiles (Figure 4). The interpretation of the seismic profiles was based on the seismic stratigraphic method developed in the mid-1970s by Mitchum and others, which distinguishes the seismic units to be interpreted based on terminations and outlines of reflections [26]. This method is standard procedure for the interpretation of all kinds of reflection seismic data from, for example, the shallow part of the seafloor e.g., [27,28], as well as for ground-penetrating radar (GPR) data.

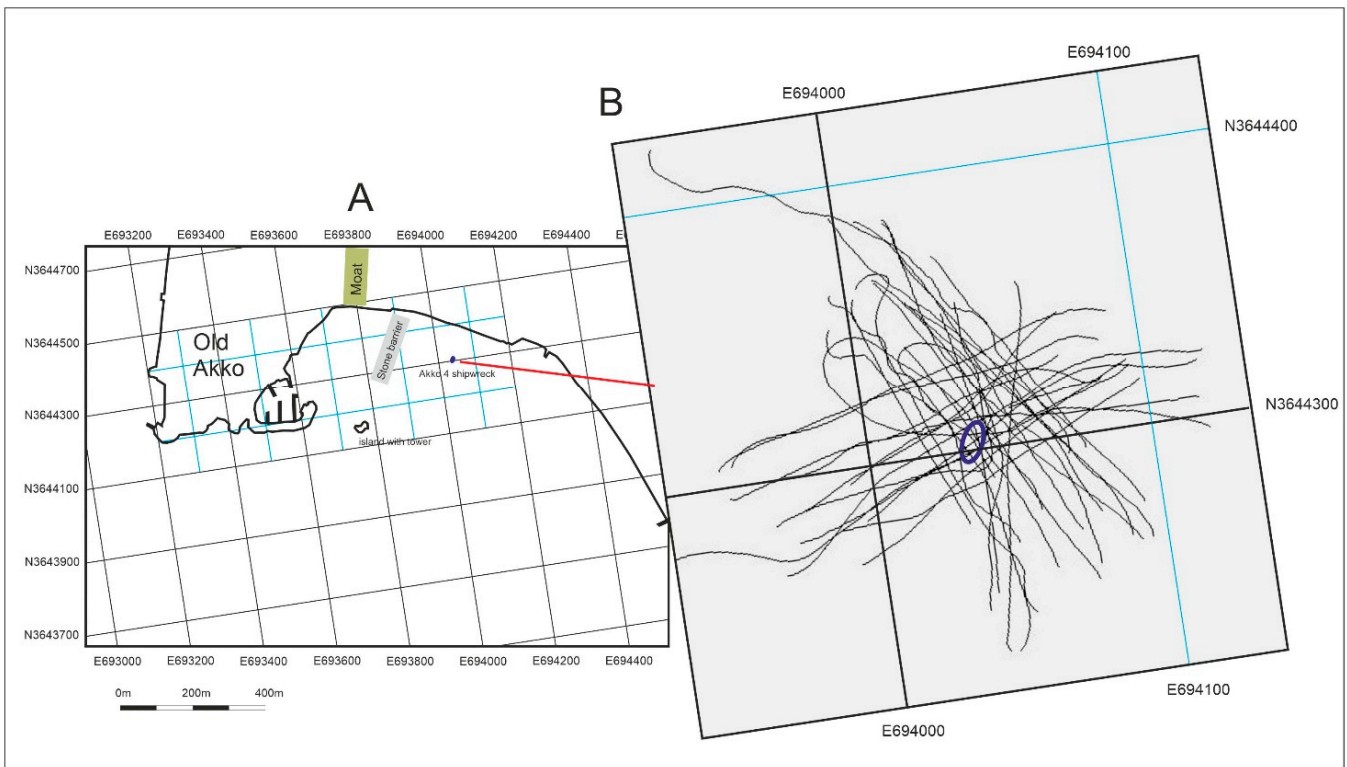

**Figure 4.** (**A**): The position of the Akko 4 shipwreck at Akko. (**B**): The Akko 4 shipwreck, marked as a blue oval with the relevant recording lines. Graphics: Grøn.

The seismic profiles from the Akko area were processed on a workstation with Petrel software, and interpreted in accordance with the seismic stratigraphic interpretation method. An area constituting a synform, a concave formation composed of sediment layers, and obviously not related to the relatively recent geological evolution of the Akko area, was interpreted as the Akko 4 shipwreck and is shown in Figure 5. This interpretation was supported by the presence at a recorded position (UTM z36 S 694040.91E 3644306.79N) inside this feature of wood and a piece of rope in a sample taken from the wreck during an earlier trial investigation (Figure 5). Figures 6–11 show clear examples of this synform, as shown on sailing recording lines 067, 008, 014, 024, and 031, marked with a dotted blue line. Above the blue line, inside the shipwreck, several prograding sand bodies can be seen to have moved northwards and eastwards (trace nos. 7070–7137, 1821–1873, 1261–1314, 1261–1338, and 1071–1007, respectively, around a depth of 3–4.5 ms TWT [two-way travel time] = 3–4.5 m below the sea surface). The synform has apparently halted their further travel, as one would expect from a physical barrier. The base of these sloping depositional surface structures ('clinoforms') lies at a depth of approximately 5 ms TWT (c. 5 m below the sea surface—trace nos. 7137–7245, 1837–1988, 1296–1468, 1338–1583, and 991–808, respectively). To the north, the reflection pattern is slightly more chaotic, with a higher amplitude ('more coloured'), thereby indicating that material much harder than sand is located close to the blue line (e.g., Figures 8 and 10, trace nos. 1225–1296 and 1088–991).

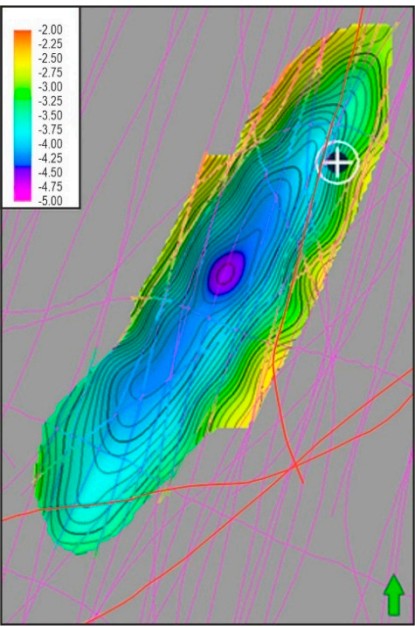

**Figure 5.** 3D reconstruction of the Akko 4 shipwreck embedded in the seafloor sediments with the recording lines shown in violet and red, and the depths estimated on the basis of a sound speed of 2000 m/s. The white cross marks the position of the previously recovered wood and rope samples. Graphics: Boldreel and Grøn.

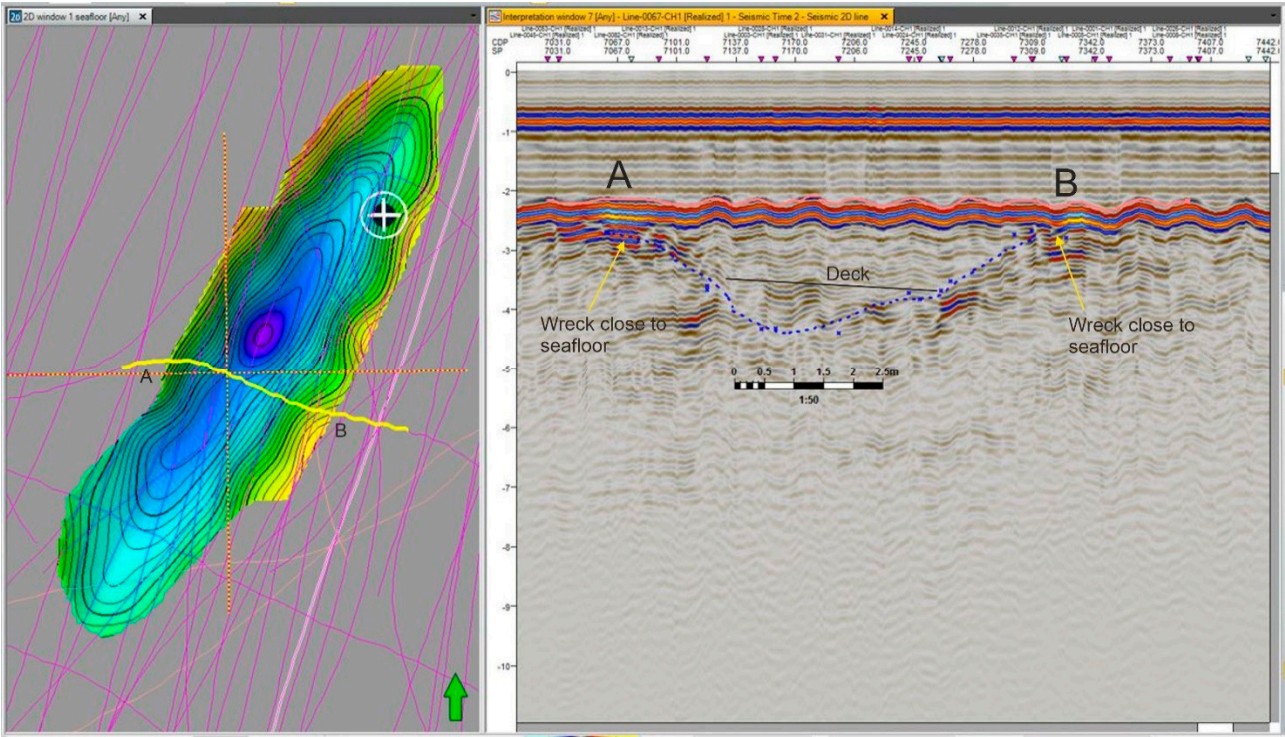

**Figure 6. Left**: Line 067 through the shipwreck. The white cross marks the position of the previously recovered wood and rope samples. **Right**: The corresponding 2D profile recorded through the wreck. The seafloor is just below the letters A and B. The dotted blue line marks the concave synform. The feature interpreted as a deck is marked with a black line. The figure's horizontal and vertical scales are identical. Small triangles at the top represent intersecting recorded Chirp profiles. The vertical scale corresponds to depth in m. Boldreel and Grøn.

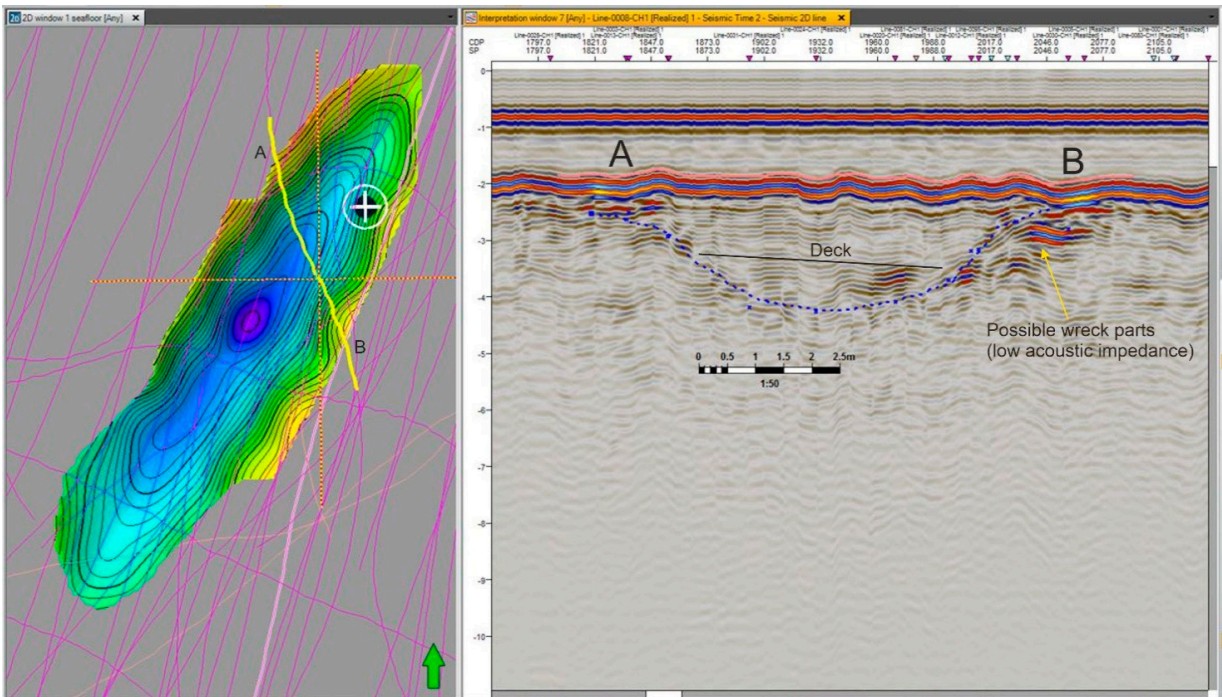

**Figure 7. Left**: Line 008 through the shipwreck. The white cross marks the position of the previously recovered wood and rope samples. **Right**: The corresponding 2D profile recorded through the wreck. The seafloor is just below the letters A and B. The dotted blue line marks the concave synform. The feature interpreted as a deck is marked with a black line. The figure's horizontal and vertical scales are identical. Small triangles at the top represent intersecting recorded Chirp profiles. The vertical scale corresponds to depth in m. Graphics: Boldreel and Grøn.

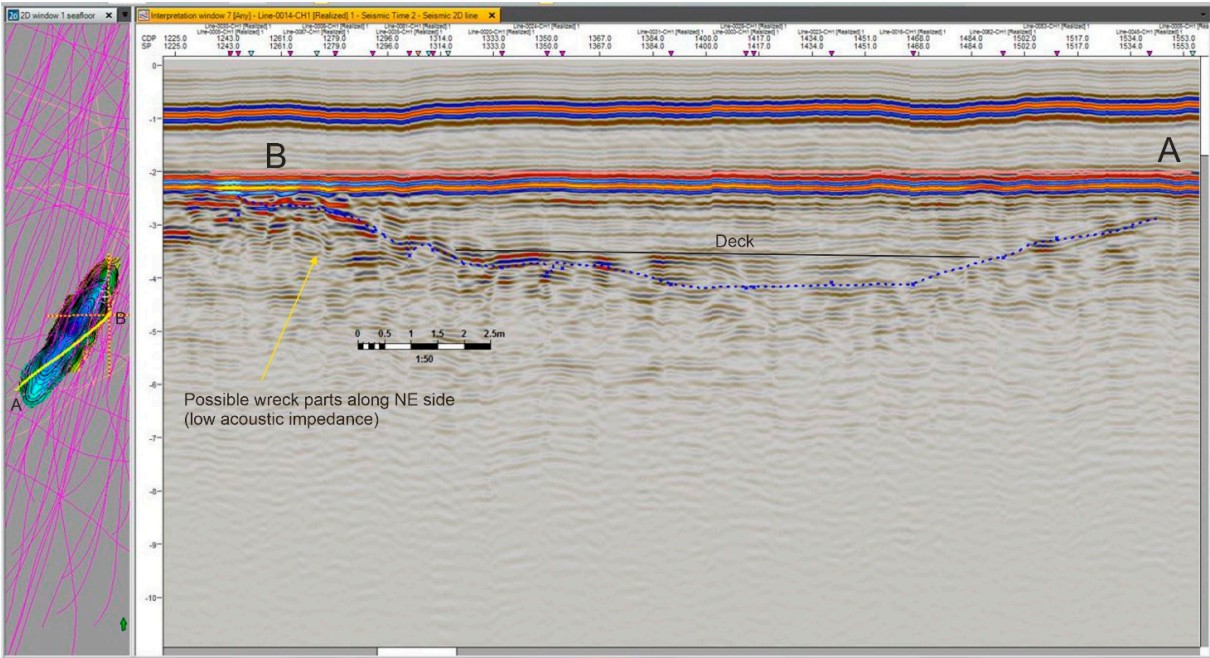

**Figure 8. Left**: Line 014 through the shipwreck. The white cross marks the position of the previously recovered wood and rope samples. **Right**: The corresponding 2D profile recorded through the wreck. The seafloor is just below the letters B and A. The dotted blue line marks the concave synform. The feature interpreted as a deck is marked with a black line. The figure's horizontal and vertical scales are identical. Small triangles at the top represent intersecting recorded Chirp profiles. The vertical scale corresponds to depth in m. Graphics: Boldreel and Grøn.

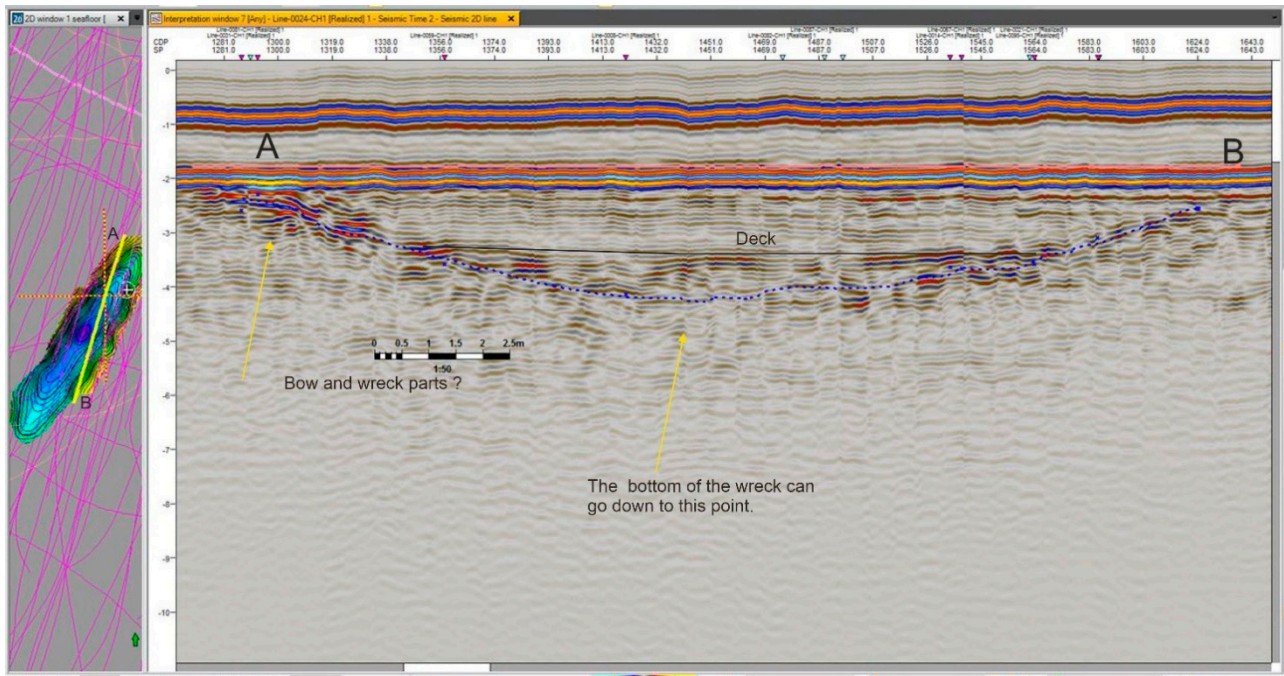

**Figure 9. Left**: Line 024 through the shipwreck. The white cross marks the position of the previously recovered sample containing wood and rope. **Right**: The corresponding 2D profile recorded through the wreck. The seafloor is just below the letters A and B. The dotted blue line marks the concave synform. The feature interpreted as a deck is marked with a black line. The figure's horizontal and vertical scales are identical. Small triangles at the top represent intersecting recorded Chirp profiles. The vertical scale corresponds to depth in m. Graphics: Boldreel and Grøn.

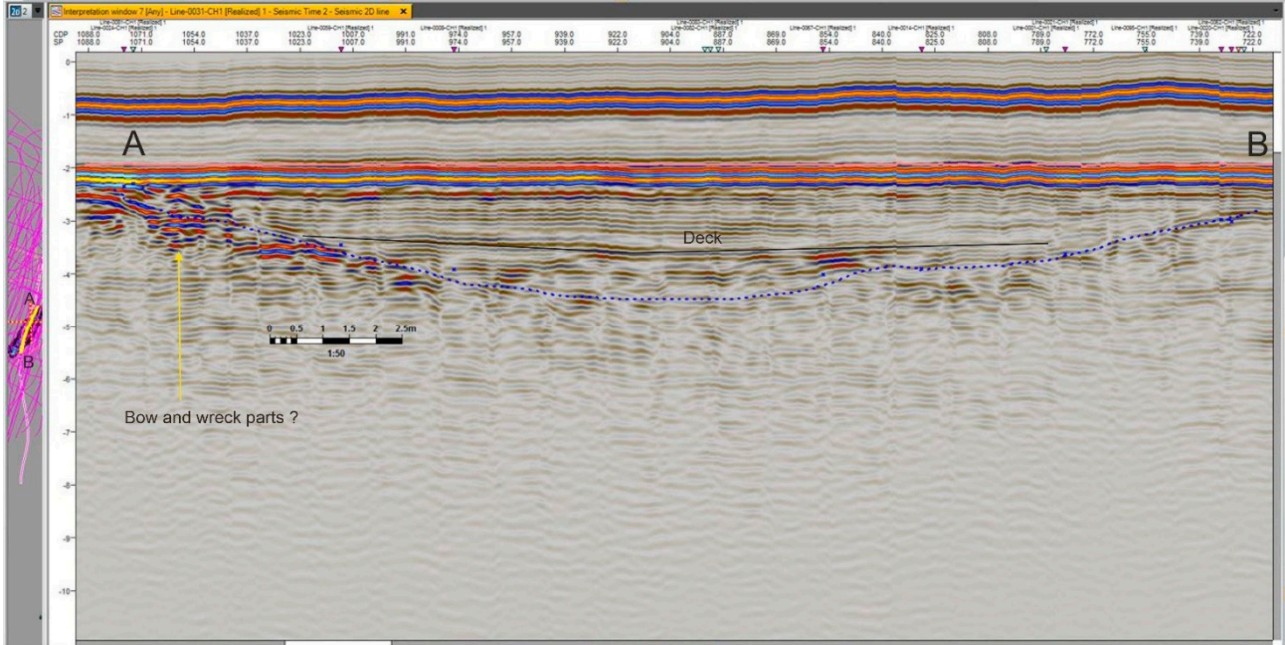

**Figure 10. Left**: Line 031 through the shipwreck. **Right**: The corresponding 2D profile recorded through the wreck. The concave synform is marked with a dotted blue line. The seafloor is just below the letters A and B. The feature interpreted as a deck is marked with a black line. The figure's horizontal and vertical scales are identical. Small triangles at the top represent intersecting recorded Chirp profiles. The vertical scale corresponds to depth in m. Graphics: Boldreel and Grøn.

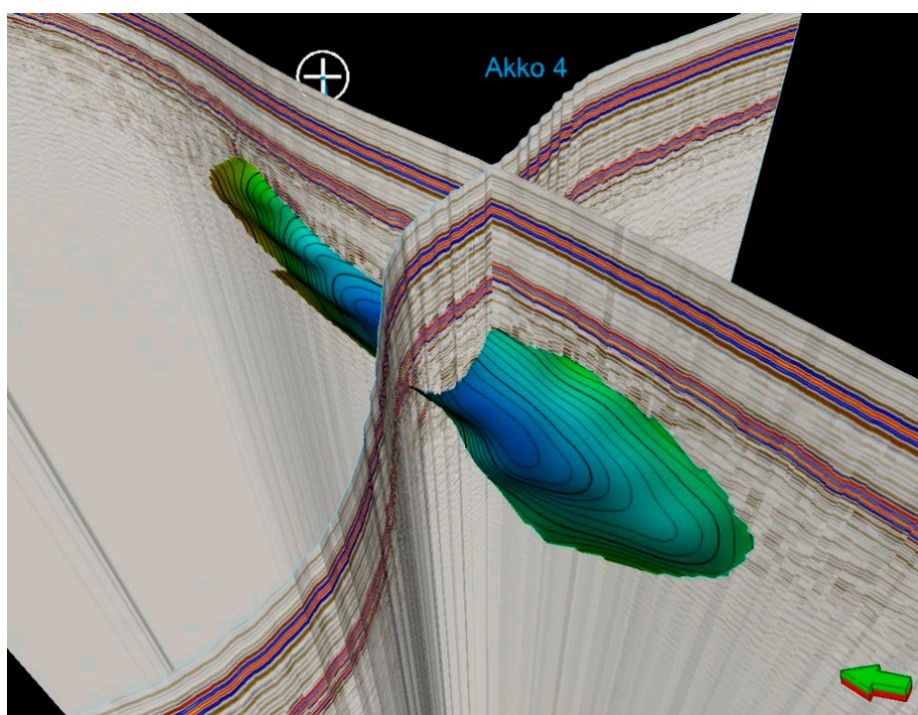

**Figure 11.** 3D reconstruction of the Akko 4 shipwreck based on all the profiles recorded with the intersecting profiles 020 and 067 shown. The white cross marks the position of the previously taken sample. Graphics: Boldreel.

The seafloor is just below the letters A and B in Figures 6–10, marking the limits of the synform. Between the seafloor and the surface of the sea is the 'top-noise', a horizontal noise band created by the seismic system itself (e.g., Figure 6, right). At the very northern part of the synform, a change in the reflection representing the seafloor is evident in Figures 6–10, between trace nos. 7067–7137, 1821–1873, 1243–1279, 1281–1319, and 1071–1037, respectively, NE of which the reflection becomes much brighter, indicating that the material changes from sand to something different, located slightly below the seafloor. The central part of the synform between the dotted blue line and the seafloor shows four horizontal seismic units with lesser infill between them (Figures 6–10, trace nos. 7101–7245, 1873–1968, 1314–1468, 1356–1564 and 991–840, respectively). Above these four patterns, a pronounced horizontal reflection connecting to the outline of the synform (the ends of the dotted blue line) is evident at both ends (trace nos. 7067–7309, 1821–2046, 1243–1534, 1281–1621, and 1071–722, respectively) at a depth of c. 3.5 ms TWT = c. 3.5 m. Above this horizontal reflection, the sedimentary infill is separated into two units.

Based on this analysis, the synform marked with the blue dotted line is interpreted as representing the hull of the wreck, as it does not match any geological structures. The hull appears to be resting on the former seafloor at a depth of c. 5 ms TWT = c. 5 m, and seems to be oriented approximately N–S and have an extent (length) of approximately 23 m at the upper part of the hull and a W–E extent (width) of approximately 7 m. At its southern end, prograding sand bodies appear to have gradually accumulated and buried the wreck. At its northern end, a harder material seems to frame this part of the hull, as the amplitude of the reflections is rather high. The upper part of the wreck here appears to be buried beneath a thin covering of sandy material on the seafloor. These features are consistent with those observed in several other profiles.

A horizontal dark band, up to about 1 m above the base of the shipwreck, which appears clearly in the profiles, cutting the structure lengthwise (Figures 8–10) is interpreted as a deck. The purpose of a deck in a merchantman would be to shield the more sensi-

tive parts of the cargo from moisture and to provide the crew somewhere to stand for manoeuvring the ship and handling the anchors.

Inside the shipwreck, below the presumed deck structure, dark shading indicates some kind of fill (Figures 6–10). This could represent ballast stones. It would, however, constitute quite an impressive amount of ballast for a vessel of this size, and experience with the applied Teledyne Chirp III suggests that a ballast deposit would appear with all the individual stones distinguishable. The most likely interpretation therefore seems to be that the fill below the deck comprises insoluble cargo.

Based on the large number of seismic profiles recorded, a 3D presentation of the wreck was reconstructed, with a white cross marking the position of the previously taken samples containing wood fragments and rope mentioned above. Based on the interpreted border of the synform (i.e., the dotted blue lines shown in Figures 6–10) a polygon was drawn around it, and a standard algorithm was used to interpolate the hull shape between the points, with the colours representing the depth below the seafloor (Figure 11).

## 4. Discussion and Conclusions

The case of Akko 4 presented here demonstrates that the application of relatively cheap off-the-shelf technology, in this case, a Teledyne Chirp III sub-bottom profiler, in combination with sufficiently precise navigation, can provide 3D reconstructions of sediment-embedded shipwrecks prior to further investigation. The quality of the digital reconstruction of the Akko 4 wreck, which is embedded in sand, is at least on a par with that of the 3D reconstruction of the *Grace Dieu* shipwreck. This is especially significant considering the fact that the recordings of the latter were targeted at this specific feature, thereby providing a much greater number of profiles running through it, and that the *Grace Dieu* was embedded in softer sediments, thereby providing a better opportunity for acoustic penetration [14]. The system used to record the *Grace Dieu* was the GeoChirp 3D from Geoacoustics, which is highly sensitive to surface dynamics (i.e., waves, wind, etc.), and must therefore be moved rather slowly—often being pushed/pulled by divers [14], which are drawbacks for the systematic survey of large areas. However, a system of this kind may be well-suited to detailed studies of known objects.

The Akko study demonstrates that useful 3D data can be obtained from buried objects already in a survey phase, by using a sub-bottom profiler that can be moved at a speed of 1–2 knots, and consequently can be used to cover larger areas. Had the number of recording lines in the earlier surveys of the Lundeborg 1 and Haithabu 4 wrecks been increased to some degree, it is likely that good 3D models could also have been produced based on these data [9]; [10]. This is an important point, as it is essential in the management of our maritime cultural heritage to develop a methodology that can be used to map sediment-embedded archaeological features that are not visible above the seafloor.

Figure 6 (Line 067) and 7 (Line 008) display minor wave disturbance—the bottom and the sub-bottom features appear wavy due to their varying distance to the surface from which the recordings were made. Technically, it is possible to compensate for the vertical wave displacement locally, as has been done for Line 067 where it cuts through the Akko 4 shipwreck (Figure 12). Figure 12 A shows the original recording, whereas Figure 12B shows the same data with wave compensation for the shipwreck area. The 'cost' of this operation is that the noise features in the water phase appear wavy, which is unimportant as it is the shipwreck which is the focus of the investigation. The improved precision of the exact depth of the shipwreck features facilitates a more precise distinction of the shape of the hull. In this case, the wave disturbance is rather unimportant. For recordings in situations with larger waves e.g., [29] wave compensation can provide an important improvement of the data before the interpretation.

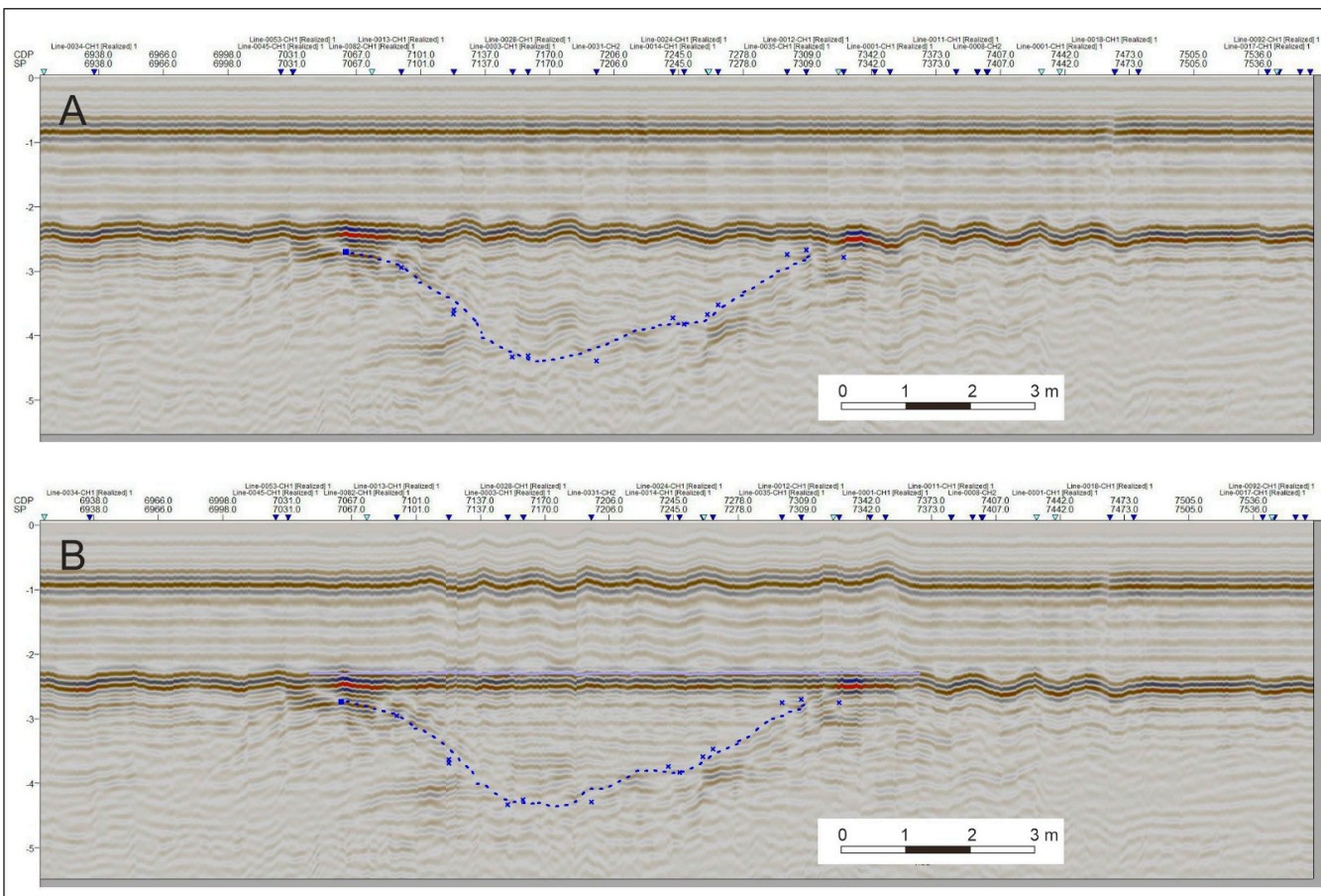

**Figure 12.** Line 067 through the Akko 4 shipwreck (see Figure 6 for location of line 067). (**A**): The originally recorded data, where the dotted blue line marks the hull. (**B**): The wave-compensated data, extent marked by the thick purple line at the seafloor, where the dotted blue line marks the hull with improved precision. The horizontal and vertical scales of the two figures are identical. Small triangles at the top represent intersecting recorded Chirp profiles. The vertical scale corresponds to depth in m. Graphics: Boldreel and Grøn.

Observations, at the time when the Akko 4 shipwreck was partially exposed, show that the lower part of the hull was covered by copper sheeting. The general experience is that metal is very difficult to observe directly in sub-bottom profiles e.g., [10]. Accordingly, no features in the recorded profiles appear to reflect the copper plating. The sub-bottom data recorded in this case not only permitted a reconstruction of the buried hull, but also provided good and consistent information about its internal features, as well as possible cargo inside the vessel's hold. Consequently, the generation of 3D images based on 2D high-resolution sub-bottom profiles with precise navigation, appears to have major potential as a basis for good strategic decision making.

**Author Contributions:** All three authors have been engaged in collecting the seismic data. L.O.B. and O.G. have interpreted the data. O.G. and D.C. have contributed to the maritime archaeology while Boldreel has contributed with geological/geophysical knowledge. All authors have read and agreed to the published version of the manuscript.

**Funding:** Israel Science Foundation (grant no. 1899/12).

**Institutional Review Board Statement:** Not relevant.

**Informed Consent Statement:** Not relevant.

**Data Availability Statement:** Not relevant.

**Acknowledgments:** The Akko 4 project was supported by the Israel Science Foundation (grant no. 1899/12), and conducted with the aid of Amir Yurman and Moshe Bachar from the maritime workshop of the Leon Recanati Institute for Maritime Studies, Haifa, and Peer T. Jørgensen, Department of Geosciences and Natural Resource Management, Copenhagen, who had the role of technician. We thank the Israel Antiquities Authority for access to the data from the 1990 survey of Akko Marina. The remote sensing survey in Akko was carried out in 1990 by the Israel Department of Antiquities and Mr Gideon Amit from the Israel Oceanographic and Limnological Research Institute. Schlumberger is thanked for the university grant issued to the Department of Geosciences and Natural Resource Management, Geology group, University of Copenhagen.

**Conflicts of Interest:** The authors declare no conflict of interest.

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
