# Peer review of "Synthetic 3D Recording of a Shipwreck Embedded in Seafloor Sediments: Distinguishing Internal Details"

_heritage, doi:10.3390/heritage4020032_

Round 1

Reviewer 1 Report

Dear respected researcher I am pleased to judge your good research There are some observations, which from my point of view only may agree or differ from the researcher's vision - however respect, appreciation, and priority remains for the researcher's vision.

1- Introduction:  It seems reasonable and the researcher dealt with many theories and corresponding research.

Address 2 and Address 3: As headlines:

  1. The Akko 4 Shipwreck and Its Context
  2. The 3D Reconstruction of the Akko 4 Shipwreck. An Inner Deck and Ballast Stones?

its content contained many pictures, starting from Picture 2 to Picture 11.

3- you did not adhere to the main titles and formats of the journal I did not find (methods, materials and experimental study)

4- The comment below the pictures is also large and somewhat lengthy, so the comment should be shortened with words less than that.

Accept my regards and appreciation

Author Response

Thanks for your review, I have replied to your comments in the attached file.

Best regards

Lars O Boldreel

Reviewer 2 Report

The article is dedicated to an alternative 2D mapping method (with off-the-shelf seismic equipment) used to derive a synthetic 3D model of a buried shipwreck (Akko 4) in Israel. The paper is well-written and structured. The method is sound, and the reported approach could enrich the ongoing methodological debates of underwater surveying. There are two minor points that I would like to mention and that authors should consider in a revision:

  • The authors emphasize several times that their equipment is relatively cheap, in comparison to other methods. Could the authors be a bit more precise with regard to this matter? Could you quantify the saving? And could you also highlight in your discussion in which methodological steps you see advantages over other established approaches? This would help the readership a bit more to understand the value of your approach.

  • The data presentation in your paper could benefit from new developments in 3D visualization. Could you at least mention in your discussion or conclusion (chapters) that the results of underwater surveys (with expert methods, such as sonar) would become clearer if 3D visualization techniques, such as immersive virtual reality, would be applied? There is a vivid debate in Geo-information Science pointing to new developments in associative and realistic presentations of 3D data. For example, these recent papers indicate these developments:

Keil, J., Edler, D., Schmitt, T., Dickmann, F. (2021). Creating Immersive Virtual Environments Based on Open Geospatial Data and Game Engines. In: KN - Journal of Cartography and Geographic Information, 71, online first: https://doi.org/10.1007/s42489-020-00069-6

Lütjens, M.; Kersten, T.P.; Dorschel, B.; Tschirschwitz, F. Virtual Reality in Cartography: Immersive 3D Visualization of the Arctic Clyde Inlet (Canada) Using Digital Elevation Models and Bathymetric Data. Multimodal Technol. Interact. 2019, 3, 9. https://doi.org/10.3390/mti3010009

Author Response

(The authors gave the same response as above.)

Reviewer 3 Report

Beyond the result, the contribution is interesting from a methodological point of view. The reconstruction of sedimented shipwrecks represents an important step, nowadays almost fundamental, before starting to excavate or simply guide a limited exploratory trench. The technology currently available is perhaps still immature to be able to return results on which to reason in detail, but it is destined to improve over time and this work is one steps that are useful for progressing with research.

Author Response

(The authors gave the same response as above.)
